# Functional Connectivity as an Index of Brain Changes Following a Unicycle Intervention: A Graph-Theoretical Network Analysis

**DOI:** 10.3390/brainsci12081092

**Published:** 2022-08-17

**Authors:** Uwe Riedmann, Andreas Fink, Bernhard Weber, Karl Koschutnig

**Affiliations:** Institute of Psychology, University of Graz, 8010 Graz, Austria

**Keywords:** graph theory, balance intervention, unicycle, functional brain network, fMRI

## Abstract

Grey matter volume reductions in the right superior temporal gyrus (rSTG) were observed in young adults who learned to ride a unicycle. As these decreases were correlated with the acquired ability in unicycling, the authors interpreted the change as a brain tissue reorganization to increase postural control’s automated and efficient coordination. The current study aims to further corroborate this interpretation by looking at changes in the functional brain network in the very same sample of participants. For this reason, we applied graph theory, a mathematics field used to study network structure functionality. Four global and two local graph-theoretical parameters were calculated to measure whole brain and rSTG specific changes in functional network activity following the three-week-unicycle training. Findings revealed that the Local Efficiency of the rSTG was significantly elevated after the intervention indicating an increase in fault tolerance of the rSTG, possibly reflecting decentralisation of rSTG specific functions to neighbouring nodes. Thus, the increased Local Efficiency may indicate increased task efficiency by decentralising the processing of functions crucial for balance.

## 1. Introduction

Recent research indicates that balance interventions result in various changes in structural and functional characteristics of the brain. This has mainly been shown in research dealing with the ageing brain. For example, the observation that balance-related abilities decrease with advancing age leading to increased risks of injury from falling, has been primarily caused by age-related brain changes [1]. With advancing age, balance control is pushed into a more cortically focused, attention-driven control [1,2,3,4].

This coincides with the fact that older people exhibit more brain activation even when just imagining standing or walking [5] and during an actual balance task [2]. Balance interventions can help older adults to improve their balance performance and decrease their risk of injury. Moreover, a decrease in brain activation is also associated with changes in balance tasks [6], indicating a direct relationship between brain activation and behavioural changes.

While showing more activation when performing balance tasks, the activation decreases in older adults (but not in younger adults) when increasing the cognitive load while balancing. This was shown by instructing participants to simultaneously perform a balance game and a serial subtraction task. The balancing game adapted to individual skills, and older adults performed an easier subtraction task to equalize the relative cognitive load. Still, the decrease in neuronal activation when performing both tasks at the same time was only shown in older adults [7]. The authors interpret this as age-related deterioration causing downgrading of neural activity.

Sehm and co-workers [8] found that a balance intervention caused grey matter changes in the hippocampus in healthy patients. In addition, performance improvements in participants who have Parkinson’s disease were correlated with grey matter changes in the bilateral anterior cingulate cortex, left inferior parietal cortex, right anterior precuneus, left ventral premotor cortex and left middle temporal gyrus. Moreover, the association between memory and balance interventions in healthy participants was later shown to persist in the behavioural dimension, in increased memory performance following a balance intervention [9]. The intervention had participants perform multiple different balance tasks twice a week for a total of 12 weeks. The training led to increases in balance performance as well as in memory and spatial cognition.

In [10] participants had to learn to ride a unicycle over a period of three weeks. Prior to and after the intervention structural and functional brain scans were obtained. The structural brain findings revealed a decrease in grey matter volume in the right superior temporal gyrus (rSTG) and a smaller cluster of reduction in the left parahippocampus. The rSTG is involved in different visuospatial processes such as spatial awareness, body positioning, the space-dependent motion of one’s body and more [11,12,13,14]. Similarly, the parahippocampus is associated with visuospatial processing along with cognitive processes [15]. Five weeks later, an additional follow-up scan showed a re-increase in the grey matter of the rSTG. The amount of grey matter reduction was also correlated with the unicycle-riding performance.

Additionally, this study also found an increase in fractional anisotropy in the right forceps major of the corpus callosum and the right corticospinal tract, along with an increase in the cortical thickness of the left superior part of the left precentral gyrus between the pre-test and the follow-up assessment [10].

The decrease in grey matter volume in the rSTG is somehow puzzling and difficult to interpret since, intuitively, an increase is commonly more likely associated with performance gains. However, as grey matter changes can be caused by several different underlying neuronal changes [16], the interpretation of the underlying cellular mechanisms resulting in the rSTG’s grey matter volume reduction is challenging. Weber et al. [10] preliminarily concluded that the observed GM decreases could probably hint at a reorganization of the brain tissue to support more automated and efficient coordination of postural control.

To further investigate this interpretation, we reanalysed the [10] dataset using graph-theoretical parameters of the functional resting-state scan. Graph-theoretic parameters quantify the qualities of networks on a local or global scale. Among other things, they can capture: (a) how efficiently a network is structured for communication (efficiency); (b) to what extent it is split into modules (modularity); as well as (c) the importance of a node to the function of the whole network (centrality). Using two local (rSTG) and four global (whole brain) parameters, we dissected regional changes coinciding with the grey matter reduction in the rSTG, while also investigating the effects of the unicycle intervention on the functional network spanning the whole brain.

## 2. Materials and Methods

### 2.1. Participants

The data used for the present graph-theoretical analyses were obtained in the study by [10]. A total of 23 right-handed participants (13 female) aged from 20 to 51 (M = 30.42, SD = 9.14) learned to ride a unicycle over three weeks. None of them had any prior experience in unicycle riding, and none had any prior diagnosed medical, neurological, or psychiatric illness. They also signed an informed consent agreement prior to their first MRI session.

### 2.2. Study Design and Procedure

A pre-post design was used to assess brain changes caused by learning unicycling. Before their first unicycle training session, all participants underwent an MRI session to assess structural data as well as functional resting-state data. The post-scan was obtained following a three-week unicycle training intervention. The scans were acquired as close to the last training session as possible. At the end of the study, participants were assessed regarding their unicycling proficiency by independent experts. For additional information about the dataset and study design, please see [10].

### 2.3. MRI Data Acquisition

MRI data were acquired using a 3T Magnetom Skyra scanner (Siemens Healthineers, Erlangen, Germany) using a 32-channel head coil. For structural images a high resolution (0.7 mm isotropic) T1-weighted MPRage-Sequence was used (TR = 2400 ms, TI = 1000 ms, TE = 2.32 ms, matrix = 320 × 320, FOV = 224 mm, 192 slices, thickness 0.7 mm, no gap, no PAT, FA = 8°) [17]. For functional data acquisition, 260 volumes (6 min 4 s) were recorded while participants were in a passive resting state. This was achieved using a T2-weighted multiband SMS-EPI-Sequence (Voxelsize = 2.5 mm isotropic, TR = 1.4 s, TE = 35 ms, matrix = 90 × 90, FoV = 225, FA = 65°).

### 2.4. Preprocessing

Results included in this manuscript come from preprocessing performed using fMRIPrep 20.1.1 (RRID:SCR_016216) [18,19], which is based on Nipype 1.5.0 (RRID:SCR_002502) [20,21]. Many internal operations of fMRIPrep use Nilearn 0.6.2 (RRID:SCR_001362) [22], mainly within the functional processing workflow.

#### 2.4.1. Anatomical Data Preprocessing

The T1-weighted images were corrected for intensity non-uniformity (INU) with N4BiasFieldCorrection [23], distributed with ANTs 2.2.0 (RRID:SCR_004757) [24]. After skull-stripping, brain tissue segmentation of cerebrospinal fluid (CSF), white matter (WM) and grey matter (GM) was performed on the brain-extracted T1w using fast (FSL 5.0.9, RRID:SCR_002823) [25]. Volume-based spatial normalization to the MNI152NLin2009cAsym standard spaces was performed through nonlinear registration with antsRegistration (ANTs 2.2.0).

#### 2.4.2. Functional Data Preprocessing

First, a reference volume and its skull-stripped version were generated using a custom methodology of fMRIPrep. Head-motion parameters with respect to the BOLD reference are estimated before any spatiotemporal filtering using mcflirt (FSL 5.0.9) [26]. BOLD runs were slice-time corrected using 3dTshift from AFNI 20160207 (RRID:SCR_005927) [27]. A deformation field to correct for susceptibility distortions was estimated based on fMRIPrep’s fieldmap-less approach. A corrected EPI (echo-planar imaging) reference was calculated for a more accurate co-registration with the anatomical reference based on the estimated susceptibility distortion. The BOLD reference was then co-registered to the T1w reference using bbregister [28]. Co-registration was configured with six degrees of freedom. Finally, the BOLD time series were resampled into the MNI152NLin2009cAsym standard space. Several confounding time series were calculated based on the preprocessed BOLD, including framewise displacement (FD) and three region-wise global signals. FD was computed using two formulations following Power (absolute sum of relative motions, [29]) and Jenkinson (relative root mean square displacement between affines, [26]). The three global signals were extracted within the CSF, the WM, and the whole-brain masks. Six motion parameters, framewise displacement, and the mean signal from CSF were regressed out of the data. Finally, smoothing with an FWHM of 2mm was performed on the data before network construction.

### 2.5. Graph Theoretical Analysis

Network construction and parameter calculation were performed using the Matlab Toolbox GRETNA v2.0 [30]. Here, we present a brief overview of our chosen nodes, edges and thresholds.

#### 2.5.1. Node Definition

Defining the nodes was achieved by partitioning the brain into 116 separate regions using the Automated Automatical Labeling atlas (AAL-116) [31]. The rSTG, which we were trying to analyze (see section Graph Theoretical Parameters), is one clearly defined node within this atlas. This enabled the possibility of qualitatively comparing structural changes in the rSTG with changes in the node’s graph parameters covering the rSTG.

#### 2.5.2. Edges

Edges were calculated using Pearson correlations. Correlating the mean time-courses of the fMRI-data of every single node with every other node established the edge weights between all 116 nodes. The weights were not binarized, and only positive edge weights were considered leading to a weighted undirected network.

#### 2.5.3. Thresholds

When constructing a functional network, noise and especially the method selection for edge calculation can overestimate the number of edges [32]. To take this into account, we used proportional thresholding. To this end, a percentage defines how many edges are used when constructing the network. This is done based on edge weights, meaning all edges that fall in the defined percentage of highest weights are used in the network, and all others are set to zero [33]. However, this leads to the additional problem of choosing an appropriate percentage.

Because a percentage can be chosen arbitrarily, the validity of the functional network parameter potentially values suffers. To counteract that, we calculated every participant’s network and corresponding parameters multiple times using multiple thresholds before determining the area under the curve (AUC) of the individual parameters and comparing group averages [33]. Thus, we studied a wide range of thresholds while not inflating the multiple comparison problem. For this study, 18 thresholds ranged from the biggest 6% of edges to the biggest 40% of edges (2% steps).

#### 2.5.4. Graph Theoretical Parameters

Graph theoretical parameters can be categorized into the network concepts they are measuring. We chose network concepts we found to be the most important and studied parameter of each category [34]. The global concepts we chose were network integration, network segregation, small-world characteristic and centrality. These concepts were measured by the respective parameters: Global Efficiency, Modularity, Small-Worldness and Hubness. Additionally, we looked at two local parameters representing node integration and node segregation, respectively, to analyse the right superior temporal gyrus (rSTG). Those were measured by Nodal Efficiency and Local Efficiency, respectively. We did not include parameters from all categories because the small-world property is not measurable on a local scale, and local centrality measures are highly correlated with local integration measures. The rSTG was used as the goal region for local analysis because there are no systematic studies on how changes in local functional graph-theoretic parameters correspond to changes in grey matter. As [10] have previously shown a decrease in grey matter volume in the rSTG of the same dataset, this analysis may help explain those finding by looking at the corresponding functional changes in parameters. Global Efficiency, Modularity, Small-Worldness, Nodal Efficiency and Local Efficiency were calculated as explained in [34].

##### Hubness

Hubness can be calculated in multiple ways [35]. Here, we defined the Degree Centrality value for every node in every threshold from each participant. Then we combined every single value from both pre and post-intervention data and calculated the mean and SD of the distribution. Lastly, we defined Hubs as nodes with a Degree Centrality value of at least one SD over the mean [36]. These categorizations were then used to compare whether the number of hubs changed between pre and post-intervention. Thus, this did not require the use of AUCs.

### 2.6. Statistical Analysis

Using a dependent t-test with the R package “rstatix” [37] all six parameters, Global Efficiency, Modularity, Small-Worldness, Hubness, Nodal Efficiency and Local Efficiency, were tested for group differences. Additionally, the AUCs for the Modularity parameter were calculated using the R package “DescTools” [38] because the deployed Matlab toolbox GRETNA did not output AUCs for the Modularity parameter.

### 2.7. Stability of Findings

Because there are no transparent best practices for atlas and threshold choices in functional network construction, we exploratively calculated the same parameters using different atlases and thresholds. Our focus was on exploring the stability of global parameters based on atlas size and network construction method. Thus, we used two spatially more refined graded atlases that were conventionalized based on functional data. However, this makes a reliable comparison of local parameter values between atlases impossible because there is no node fully representing the rSTG. Further, Nodes that are partially represented by voxels in the rSTG are not bound to have all their respective voxels in close vicinity of the rSTG. To explore whether a node from a functional atlas that is close to our original rSTG node has enough crossover to replicate the findings, we chose to use the node closest to our original rSTG node, based on MNI coordinates, as a representative.

For the alternative atlases, we used the Dosenbach-160 [39] and the Power-264 [40], dividing the brain into 160 and 264 segments, respectively. As for the thresholds, we used the original 18 thresholds and used the lower half (6% to 22% with 2% steps) and the higher half (24% to 40% with 2% steps) individually for all three atlases.

## 3. Results

### 3.1. Descriptive Statistics

First, we looked at the Small-Worldness measure of each constructed network to validate whether our choice in thresholds led to realistic network structures. Networks with a Small-Worldness value higher than one are considered networks with small-world properties [35]. In our analysis, only ten (1.2%) of the 828 constructed networks (23 persons × 2-time points × 18 thresholds) did not achieve a value of at least 1, showing that the chosen network construction parameters led to an overall reliable network structure.

### 3.2. Intervention Effects

All subsequently presented effects are calculated from parameters based on the AAL atlas network [31] and are Bonferroni corrected if not noted otherwise. Group differences are also graphed in Figure 1, visualizing mean parameter values and standard deviation across thresholds for every parameter except Hubness. The curves are a visual representation of the calculated AUCs used for group comparison. They show that some differences in pre- and post-parameter values are dependent on the used threshold while others seem to be stable over the whole threshold range.

#### 3.2.1. Local Effects

Learning to ride a unicycle caused a significant change in Local Efficiency in the rSTG (*t*(22) = 3.51, *p* = 0.002, *d* = 0.886). However, Nodal Efficiency did not show a significant difference between pre- and post-intervention (*t*(22) = 0.56, *p* = 0.582, *d* = 0.361) (Figure 1).

#### 3.2.2. Global Effects

After correcting for multiple comparisons, Global Efficiency did not significantly change between pre- and post-intervention (*t*(22) = 1.24, *p* = 0.226, *d* = 0.461). It is worth mentioning that without correction, the change in Global Efficiency was significant (*t_uncorrected_*(22) = 2.21, *p_uncorrected_* = 0.038). Changes in Modularity (*t*(22) = 0.00, *p* = 1.000, *d* = −0.122), Small-Worldness (*t*(22) = 0.00, *p* = 1.000, *d* = −0.176) and Hubness (*t*(22) = 0.82, *p* = 0.422, *d* = 0.397) were not significant either (Figure 1).

### 3.3. Stability of Findings

The stability of the findings is qualitatively expressed by the difference in t-values a parameter has across different thresholds and atlases. As seen in Table 1, most of the global parameters (Global Efficiency, Small-Worldness and Modularity) showed relatively stable results across different thresholds and moderately stable results across atlases. For example, Global Efficiency t-values across AAL-116 thresholds lay between 1.93 and 2.28. As for t-values of different atlases using the total threshold, the values were between 1.64 and 2.21, which we classified as close given the big differences in atlas construction.

However, Hubness and the two local parameters (Local Efficiency and Nodal Efficiency) show large differences between atlases and, in the case of Hubness, the thresholds (AAL-116, Hubness t-value range: 0.53–2.02). See Discussion for further information and Table 1 for a list of all results.

## 4. Discussion

This study is among the first to look at changes in functional brain networks following a complex balance intervention. We have shown that local network changes coincide with the reported grey matter changes in the rSTG [10]. Specifically, the Local Efficiency was higher after the intervention, indicating that the rSTG’s neighbourhood nodes (nodes of the network directly connected to the rSTG) increased their efficiency in communicating with each other. This can also be interpreted as an increase in fault tolerance because it indicates how efficiently the neighbourhood of the rSTG can communicate in the case that the rSTG is disrupted [42,43]. Interestingly, none of the global parameters significantly changed following the intervention.

The presented findings show parallels to a study by [44], which investigated the effects of aerobic training on the brain network. Similar to this study, the training significantly modulated only local parameters. These findings could indicate that local parameters are better suited to identify exercise-driven brain changes, possible because these changes are primarily driven by functional specialization [45]; of course, it may also well be the case that a dataset involving larger samples of participants is needed to detect more global brain changes.

### 4.1. Implications

First and foremost, combining this study’s findings with the results reported in [10] gives us new insights into the changes that may occur in the rSTG when learning to ride a unicycle and possibly following balance interventions as a whole. Of course, this study does not answer which micro-scale changes occur when we measure changes in grey matter. There are multiple candidates; most likely, there may be combinations of all types of cellular changes [16]. However, our findings indicate that the reduced grey matter volume may be caused, or at least be associated with increases in the interconnectedness of topologically neighbouring nodes, which [43] was also interpreted as a local increase in fault tolerance of a node. This interpretation may provide insight into why the grey matter decreased.

The heightened interconnectedness in the resting state condition may indicate the decentralisation of functions prior mainly managed by the rSTG, such as spatial awareness. It is not uncommon for structurally different parts to be able to fulfil the same function [46,47]. Thus, a higher priority of the rSTG functions stimulated through unicycle training may have led to the decentralisation, indicated by the increase in interconnectedness, which in turn led to a decrease in the grey matter of the rSTG. While only speculative, testing this hypothesis is possible [47] and could significantly increase our understanding of neuroplasticity and brain network architecture.

An additional implication is that graph-theoretic network analysis is a viable tool to study the effects balance interventions have on the brain. This study also supports the promising notion that graph-theoretical parameters may be used as markers of optimal intervention choice [48].

Future studies should focus on using a mixture of neuroimaging tools and transcranial magnetic stimulation (TMS) to validate our interpretation of the functional changes [47]. The utilization of alternative balance training modalities will also show whether the findings are unicycle training specific or generalisable to all balance interventions.

### 4.2. Stability of Findings

The comparison of results between the different atlas and threshold choices is not standard practice, but given the lack of consensus in choosing these parameters, we made great efforts to demonstrate the reliability of our findings using this method. Papers that looked at how different analytic choices affect graph-theoretical parameters primarily focused on preprocessing [49] or structural networks [50]. That being said, previous studies took a similar route to test the reliability of their findings [51].

The findings showed that Global Efficiency, Small-Worldness and Modularity measures were relatively stable across different atlases and thresholds, while Hubness and the local parameter measures were more affected by the specific choices. As mentioned in the Methods section, this was expected for the local measurements because the nodes used to represent the rSTG do not actually cover the rSTG anatomically, as the alternative atlases were not constructed based on anatomical data. Therefore, this not only shows the importance of node choice in local network analyses but also validates our choice to use an anatomical atlas for the analysis.

The Hubness is the only parameter that shows variability in the results depending on atlas and threshold choices. This is most likely caused by the complex manner the hubs are calculated. For example, calculating a cutoff point for what defines a hub is especially affected by the number of regions the atlas has and the number of thresholds used. This increases the discrepancy between the calculated thresholds far more than for the other global parameters relying on AUCs. The question of how to define hubs reliably is not a new problem [36,52], but this study underlines the importance of cautious calculation and interpretation of results when working with Hubness.

### 4.3. Methodological Limitations

This study is mainly limited by the lack of methodological consensus. The most important is that atlases based on anatomical data are not considered optimal for network construction. Even more, pre-conceptualized atlases themselves are in some ways inferior to brain parcellations that were specifically produced for the respective sample [53,54]. On the other hand, we hypothesized about structural findings in the rSTG, making node definition only possible by using an anatomical parcellation. Thus, choosing a structural atlas was not avoidable.

There is also much room for discussion that there may not be an optimal atlas, optimal parameter values and optimal preprocessing steps for network construction in the first place [55,56]. Generally, this does not invalidate the findings but it emphasizes the importance of further research seeking more reliable scientific standards for brain network analyses and, respectively, graph theoretical network analysis.

## Figures and Tables

**Figure 1 brainsci-12-01092-f001:**
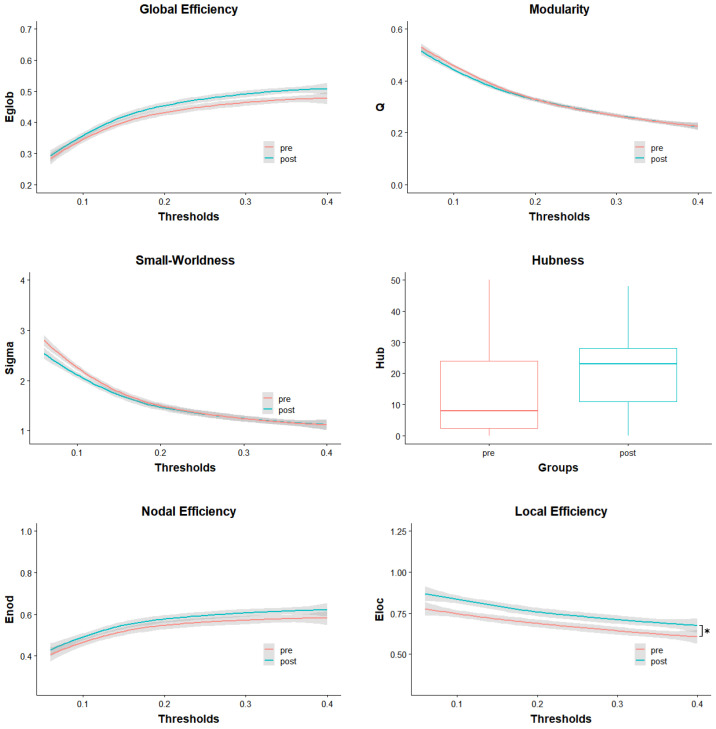
Pre- and post-test distribution of parameters over the full range of used thresholds. The grey areas indicate the standard deviations. In the case of Hubness, the groups are represented in a boxplot. Enod = Nodal Efficiency; Eloc = Local Efficiency; Eglob = Global Efficiency; Q = Modularity; Sigma = Small-Worldness; Hub = number of Hubs; * = significant difference.

**Table 1 brainsci-12-01092-t001:** Overview of all uncorrected Results.

	Total		Low		High	
	t	*p*	t	*p*	t	*p*
**AAL-116**						
Eloc	**4.25**	<**0.001**	**4.38**	<**0.001**	**3.96**	<**0.001**
Enod	1.73	0.097	1.73	0.099	1.72	0.099
Eglob	**2.21**	**0.038**	1.93	0.067	**2.28**	**0.033**
Mod	−0.59	0.564	−1.09	0.286	0.06	0.956
SW	−0.84	0.409	−1.06	0.302	0.07	0.943
Hub	1.90	0.070	2.02	0.056	0.53	0.599
**Dosenbach-160**						
Eloc	0.69	0.500	0.29	0.777	1.10	0.283
Enod	0.44	0.668	0.50	0.622	0.38	0.712
Eglob	1.83	0.080	1.79	0.088	1.51	0.146
Mod	0.03	0.979	0.20	0.847	−0.18	0.856
SW	0.01	0.991	0.01	0.992	0.03	0.978
Hub	0.39	0.702	0.28	0.783	0.44	0.666
**Power-264**						
Eloc	0.04	0.971	0.44	0.665	**136**	**0.964**
Enod	0.38	0.707	0.32	0.749	0.45	0.657
Eglob	1.64	0.115	1.70	0.104	1.46	0.159
Mod	−0.10	0.920	<0.01	0.997	−0.24	0.813
SW	−0.41	0.690	−0.40	0.690	−0.38	0.710
Hub	0.77	0.449	0.74	0.465	**108.5**	**0.378**

**Bold** = *p* < 0.05; **Italic** = Wilcoxon-Signed-Rank Test; Total = 6–40% thresholds in 2% steps; Low = 6–22%
thresholds in 2% steps; High = 24–40% thresholds in 2% steps; Eloc = Local Efficiency; Enod = Nodal Efficiency;
Eglob = Global Efficiency; Mod = Modularity; SW = Small-Worldness; Hub = Hubness. Note. The Wilcoxon-
Signed-Rank test [41] has been used in two of the presented comparisons. This was because, in both cases, the
assumptions were not met. Therefore, their values are not directly comparable to the other tests and should be
neglected when comparing outcomes.

## Data Availability

Data will be available upon request from the corresponding author.

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
