# Peer review of "Functional Connectivity as an Index of Brain Changes Following a Unicycle Intervention: A Graph-Theoretical Network Analysis"

_brainsci, 2022, doi:10.3390/brainsci12081092_

Round 1

Reviewer 1 Report

This is a very well-written manuscript, with all necessary information surrounding the study included (especially within the Methods section). With minor additions, I believe this paper is ready for publication.

1. Add significance indicators to Figure 1, showing that Local Efficiency was significantly changed (for example). This will help that measure stand out in that figure.

2. Potential further directions- if this study does have implications for "balance interventions", what can you do next?

Author Response

We would like to thank the reviewer for the prompt and very positive feedback.

Ad 1) Significant differences between pre and post are now marked with an (*) in figure 1. Thank you for this suggestion.

Ad 2) A short paragraph regarding future directions has been added to the manuscript (4.1 Implications).

Reviewer 2 Report

1. The first line in the abstract can be rewritten as, "Grey matter volume reductions in the right superior temporal gyrus (rSTG) were observed in young adults who learned to ride a unicycle.

2. I would recommend the authors to start a sentence with a word instead of a reference number. Example, in the introduction, the third paragraph should start with" Sehm etal or Sehm and coworkers found that..." 

3. Please use past tense throughout the manuscript.

For example - In "2.5.1. Node Definition" - the grammar should be consistent, so please change the second line, "The rSTG, which we are 139 trying to analyze..." to The rSTG, which we were trying to analyze. Likewise, in many other places, grammar should be kept consistent. 

4. Please check for spellings. 

Example: Section  4.3- This study is mainly, not "manly"

Author Response

We would like to thank the reviewer for the prompt and very positive feedback. All changes are marked in the pdf-version of the manuscript.

Ad 1/2) Thank you for this suggestion. We realized this (odd-looking start) after uploading the latex-file. The sentence is now rewritten.

Ad 3) Thank you again. We updated the manuscript to past tense.

Ad 4) We thoroughly checked the complete manuscript for spellings and corrected it accordingly.